# Tracking Respiratory Viruses in Pakistan (2022–2025): Genomic and Epidemiologic Insights from Sentinel Surveillance of Influenza, SARS-CoV-2, and RSV

**DOI:** 10.3390/v18010026

**Published:** 2025-12-23

**Authors:** Nazish Badar, Abdul Ahad, Hamza Ahmed Mirza, Fazal Hanan, Asghar Javaid, Aamir Amanat, Zunira Saghir, Muhammad Salman

**Affiliations:** 1Public Health Laboratories Division, National Institute of Health, Chak Shahzad, Park Road, Islamabad 45500, Pakistan; abdul.ahad@nih.org.pk (A.A.); hamzaahmadmirza@gmail.com (H.A.M.); aamiramanat57@gmail.com (A.A.); zunirasaghir53@gmail.com (Z.S.); 2Saidu Group of Teaching Hospital, Pathology Department, Saidu Medical College, Swat 19200, Pakistan; drfhanan@gmail.com; 3Pathology Department, Nishtar Medical University, Multan 66000, Pakistan; asgharmicro@hotmail.com; 4National Institute of Health, Chak Shahzad, Park Road, Islamabad 45500, Pakistan; salman14m@gmail.com

**Keywords:** respiratory viruses, sentinel surveillance, Pakistan

## Abstract

Influenza and other respiratory viruses pose significant public health threats among SARI patients, yet comprehensive surveillance data remain limited in Pakistan. This prospective, multi-center study characterized the burden, distribution, and molecular evolution of respiratory viruses among hospitalized SARI patients across seven tertiary hospitals from November 2022 to June 2025. Specimens were tested using RT-PCR for influenza, SARS-CoV-2, and RSV, with 375 samples sequenced via Oxford Nanopore Technology. Among 11,451 specimens, 2818 (24.6%) tested positive: RSV (1648, 14.4%), influenza (855, 7.5%; 45% H1N1pdm09, 35% H3N2, 20% influenza B), and SARS-CoV-2 (315, 2.8%). RSV predominantly affected children under 2 years (63%), while influenza and SARS-CoV-2 primarily impacted adults aged 15–40 years. Male predominance (65–79%) reflected healthcare access barriers. Strong winter seasonality (December–February) was observed for influenza and RSV. ICU admission rates were 17% for influenza, 16% for RSV, and 4% for SARS-CoV-2. Shortness of breath was associated with influenza (OR = 1.62) and RSV (OR = 1.27), while malaise (OR = 2.24) and myalgia (OR = 3.87) was associated with SARS-CoV-2. Phylogenetic analysis revealed vaccine-matched influenza clades and rapid SARS-CoV-2 variant succession (3–4 months). RSV is the primary SARI pathogen in young children, necessitating maternal vaccines and nirsevimab implementation. Sustained genomic surveillance remains essential for pandemic preparedness.

## 1. Introduction

Severe acute respiratory infections (SARIs) caused by influenza and other respiratory viruses (ORVs) remain a significant global public health concern, contributing to high morbidity and mortality [1]. Lower respiratory infections continue to rank among the leading causes of death worldwide and are the second-highest cause of mortality in lower–middle-income countries [2]. To address this burden, many countries have adopted sentinel surveillance systems to generate evidence on the epidemiology of influenza and ORVs. However, data gaps remain, particularly in low- and middle-income countries [3]. The global focus on SARS-CoV-2 diverted resources and attention from other respiratory pathogens, limiting systematic screening and masking the actual burden of influenza and ORVs [4]. As the world transitions into the endemic phase, strengthening integrated surveillance of SARI has become critical. Such systems are essential to assess shifts in disease epidemiology, seasonal patterns, clinical outcomes, healthcare utilization, and the emergence of new viral variants, while also informing preparedness for future outbreaks [5]. The recent availability of new preventive interventions, including maternal respiratory syncytial virus (RSV) vaccines and long-acting monoclonal antibodies, further underscores the need for robust surveillance data to guide implementation strategies [6].

In recognition of these needs, Pakistan joined the Global Influenza Hospital-based Surveillance Network (GIHSN) in 2022. This collaboration marked an important step toward building a standardized, multi-site sentinel surveillance system for influenza and ORVs. By generating harmonized data, the GIHSN platform strengthens understanding of seasonal patterns, clinical and genomic characteristics, and disease burden across diverse populations and geographies. Pakistan’s large population (>240 million), diverse climatic zones ranging from tropical to temperate, and strategic geographic location at the crossroads of South and Central Asia make robust respiratory virus surveillance particularly relevant for both regional and global health security [7].

Despite Pakistan’s participation in global surveillance networks, comprehensive data characterizing the full spectrum of respiratory viruses causing SARI, their age-specific distribution, seasonal patterns, clinical severity, and molecular evolution remain limited [8]. Most previous surveillance efforts focused primarily on influenza, with limited systematic assessment of RSV, SARS-CoV-2 in the endemic phase, and other respiratory pathogens [9,10]. Understanding these dynamics is essential for guiding evidence-based interventions, including vaccination strategies, clinical management protocols, and resource allocation for pandemic preparedness [11].

This study characterizes the burden, distribution, and molecular evolution of influenza, SARS-CoV-2, RSV, and other ORVs through integrated sentinel surveillance of hospitalized SARI cases across seven tertiary care hospitals in Pakistan from November 2022 to June 2025. Through universal testing for major respiratory pathogens and whole-genome sequencing of 375 viral samples, we provide comprehensive insights into clinical and epidemiological burden, pathogen distribution, age-specific vulnerabilities, seasonal transmission patterns, and molecular evolution of circulating respiratory viruses. These findings aim to inform national prevention strategies, strengthen healthcare preparedness, and contribute to global understanding of respiratory virus epidemiology in South Asia.

## 2. Materials and Methods

### 2.1. Study Design and Setting

This prospective, multi-center sentinel surveillance study was conducted from 1 November 2022, to 30 June 2025 (31 months), across seven strategically chosen tertiary care hospitals in Pakistan. The study used an inferential cross-sectional design to thoroughly characterize the epidemiology, clinical features, and molecular evolution of respiratory viruses causing SARI in hospitalized patients.

The surveillance network was coordinated by the National Influenza Center (NIC) at the National Institute of Health (NIH), Islamabad, which served as the central reference laboratory and coordinating hub. The seven sentinel sites were selected across various regions of Pakistan, including major urban population centers from four provinces and two administrative areas, as shown in Figure 1. At each sentinel site, dedicated surveillance teams were established, comprising clinical coordinators from the Departments of Medicine, Pediatrics, Ear, Nose, and Throat, and Obstetrics and Gynecology; trained laboratory technicians responsible for sample collection, processing, and shipment; and data entry personnel.

### 2.2. Case Definition and Patient Enrollment

Patients were enrolled based on the WHO criteria for SARI: acute respiratory illness with fever ≥ 38 °C, cough, and hospital admission. Suspected cases were assessed by surveillance technicians and enrolled after informed consent [12]. Demographic data (age, sex, location), clinical presentation (symptoms, onset date), and severity indicators (oxygen requirement, ICU admission, mechanical ventilation, High Dependency Unit (HDU) admission) were collected using standardized case report forms, with data cross-verified with medical records.

### 2.3. Specimen Collection and Laboratory Testing

Nasopharyngeal and throat swabs were collected within 24–48 h of admission using standard methods and stored at −70 °C until testing. Total nucleic acid was extracted from 200 μL of clinical specimens using the KingFisher™ Flex Purification System with the MagMAX™ Viral/Pathogen Nucleic Acid Isolation Kit (ThermoFisher Scientific, Waltham, MA, USA). All samples underwent universal testing for influenza A, influenza B, and SARS-CoV-2 with the CDC Influenza SARS-CoV-2 (Flu SC2) Multiplex Assay (FluSC2PPB-RUO, Centers for Disease Control and Prevention, Atlanta, GA, USA) on the Applied Biosystems 7500 Fast Real-Time PCR System. Influenza A-positive samples were subtyped using CDC Influenza A (H3/H1pdm09) Subtyping Panel (Catalog # FluRUO-16, CDC Influenza Division, Atlanta, GA, USA) with TaqPath™ RT-PCR kit (ThermoFisher Scientific, Waltham, MA, USA). RSV detection was performed using the CDC RSV Multiplex Assay (GR-1365). A representative subset of 10% of specimens testing negative for influenza, RSV, and SARS-CoV-2 underwent expanded multiplex testing using the Fast Track Diagnostics (FTD) Respiratory Pathogens 21 Kit (Siemens Healthineers, Erlangen, Germany), which detects rhinovirus, human metapneumovirus (hMPV), adenovirus, parainfluenza viruses (types 1–4), human coronaviruses (229E, NL63, OC43, HKU1), bocavirus, enterovirus, and parechovirus [13].

Sample selection employed a stratified approach prioritizing: (1) ICU-admitted or mechanically ventilated patients, (2) cases with severe unexplained symptoms, and (3) stratified random sampling ensuring geographic and temporal representation across all sites and surveillance periods. This strategy maximized pathogen detection within resource constraints but introduced ascertainment bias by enriching for severe cases. Consequently, prevalence estimates and severity indicators for these additional respiratory viruses (rhinovirus, hMPV, adenovirus, parainfluenza) likely overestimate their true burden among all SARI patients and should be interpreted with this limitation in mind.

### 2.4. Whole-Genome Sequencing and Bioinformatics Analysis

Specimens with sufficient viral load were selected for whole-genome sequencing. For SARS-CoV-2, samples with Ct values < 28 were prioritized to ensure adequate viral genome coverage and sequencing quality. For influenza viruses, samples with Ct values < 30 and adequate RNA quality (assessed by RNA integrity and concentration) were selected, as influenza genome amplification protocols are optimized for this threshold. The slightly higher Ct threshold for influenza reflects the multi-segment nature of the influenza genome and the need for balanced amplification across all eight segments [14].

### 2.5. Influenza Virus Sequencing

For influenza A virus sequencing, the Oxford Nanopore platform was used following the protocol described by [15]. For influenza B virus, the protocol described by [16] was followed. Basecalling and demultiplexing of POD5 files were performed using Dorado (v1.2.0), followed by removal of low-quality reads (<300 bp or >3000 bp) using SeqKit (v2.10.0) and adapter trimming with Porechop (v4.9.1). Influenza virus genome segments were analyzed independently by mapping demultiplexed reads to segment-specific reference sequences using Minimap2 v2.24. Reference strains used were: A/Wisconsin/588/2019 (H1N1pdm09), A/Darwin/6/2021 (H3N2), and B/Austria/1359417/2021 (Influenza B). Variant calling was performed using SAMtools (v.1.21) and BCFtools (v.1.18), with a 10× coverage threshold applied and segments excluded if they had less than 80% coverage. Data visualization and quality assessment were carried out using Qualimap v2.3 and IGV v2.17.4.

### 2.6. SARS-CoV-2 Sequencing

For SARS-CoV-2 sequencing and bioinformatics, the protocol described by [9] was followed. Briefly, consensus genomes were generated using a Nanopolish workflow based on the modified ARTIC pipeline (v1.8.5), with reads aligned to the Wuhan-Hu-1 reference genome (GenBank accession MN908947.3). Variant polishing was performed with Nanopolish (v0.8.4), Medaka (v1.12.1), and Samtools (v1.21).

### 2.7. Phylogenetic Analysis

Phylogenetic trees were created using closely related sequences retrieved from GISAID on 5 July 2025 (https://gisaid.org/). For each virus, representative Pakistani sequences were selected along with contemporaneous global sequences to provide geographic and temporal context. Substitution models were predicted with jModelTest and multiple sequence alignment was performed with MAFFT. Maximum likelihood (ML) phylogenetic trees were generated using IQ-TREE. Trees were rooted with vaccine reference strains: A/Darwin/6/2021 for influenza A(H3N2), A/Wisconsin/588/2019 for influenza A(H1N1)pdm09, B/Austria/1359417/2021 for influenza B, and Wuhan-Hu-1 for SARS-CoV-2. Tree visualization and editing were conducted in FigTree v1.4.4. Clade and subclade classification for Influenza and SARS-CoV-2 was performed using Nextstrain and nextclade v.3.18.1. For temporal tracking of variant succession, Pango lineage designations were assigned to all 148 sequenced genomes using pangolin [17].

### 2.8. Statistical Analysis

Statistical analyses were performed using SPSS version 28.0 and R version 4.3.1. Descriptive statistics were calculated for demographic and clinical variables. Categorical variables were compared using Pearson’s χ^2^ test or Fisher’s exact test where appropriate. The prevalence and distribution of respiratory viruses were analyzed using χ^2^ tests to identify significant deviations from expected patterns. Monthly variations in positive cases were assessed to detect seasonal patterns, using χ^2^ tests to compare observed and expected frequencies. The relationship between age groups and virus types was examined using χ^2^ tests to determine whether certain viruses disproportionately affected specific age demographics.

Binary logistic regression was performed to identify factors associated with infection by each virus type (influenza, RSV, SARS-CoV-2). Initially, univariable logistic regression analyses were performed for demographic variables (age and sex) and clinical symptoms (fever, cough, sore throat, shortness of breath, wheezing, nasal congestion, malaise, headache, and myalgia). Variables for the multivariable logistic regression were selected based on bivariable associations (*p* ≤ 0.20) and clinical relevance. Confounders, such as age, sex, and comorbidities, were included based on the literature and conceptual frameworks. For each virus, a separate multivariable model was constructed. Results are reported as odds ratios (OR) with 95% confidence intervals (CI). *p*-values < 0.05 were considered statistically significant.

Model diagnostics included assessment of collinearity using variance inflation factor (VIF), evaluation of linearity of the logit, goodness-of-fit testing using the Hosmer–Lemeshow test, discrimination assessment using receiver operating characteristic curve analysis (ROC) with area under the curve (AUC), and evaluation of influential observations. Internal validation was performed using bootstrapping with 1000 resamples. Forest plots were generated to visualize adjusted odds ratios and 95% confidence intervals from multivariable models using the forestplot and ggplot2 packages in R [18].

## 3. Results

During the study period, a total of 11,451 samples were tested for influenza and other respiratory viruses. Of these, 2818 (24.6%) tested positive for respiratory pathogens: influenza (855 cases, 7.5%), RSV (1648 cases, 14.4%), and SARS-CoV-2 (315 cases, 2.8%). As part of expanded pathogen surveillance efforts, a representative subset of 870 (10%) from the 8633 negative samples was selected for multiplex testing. This expanded testing identified rhinovirus in 115 samples (1%), human metapneumovirus in 87 samples (0.76%), adenovirus in 49 samples (0.43%), and parainfluenza virus in 33 samples (0.29%). Demographic and clinical characteristics of patients stratified by viral etiology are presented in Table 1.

### 3.1. Demographic and Temporal Distribution of Respiratory Viruses

Among all 11,451 cases, 7486 (65%) were male and 3965 (35%) were female. Male predominance was observed across all pathogens: influenza (79% vs. 21%), RSV (76% vs. 24%), SARS-CoV-2 (65% vs. 35%), and ORVs (69% vs. 31%).

Age distribution varied significantly by pathogen type. Overall, 3312 cases (29%) were children under 2 years, 459 (4%) were aged 2–4 years, 873 (8%) were aged 5–14 years, 2749 (24%) were aged 15–40 years, 2519 (22%) were aged 41–64 years, and 1539 (13%) were 65 years or older. RSV predominantly affected young children, with 63% (1039/1648) of RSV cases occurring in children under 2 years. In contrast, influenza and SARS-CoV-2 were more common in adults aged 15–40 years, accounting for 35% (302/855) and 22% (68/315) of cases, respectively (Figure 2).

Figure 3 illustrates the demographic distribution of respiratory virus cases across age groups and gender. Children under 2 years constituted the most vulnerable demographic, accounting for 29% of all tested cases (3312/11,451) and representing 63% of RSV cases, 34% of influenza cases, and 13% of SARS-CoV-2 cases. Adults aged 15–40 years comprised 24% of all cases and represented the primary age group for influenza (35%) and SARS-CoV-2 (22%), while accounting for only 9% of RSV cases. The 41–64-year age group showed steady case numbers (22% of all cases, n = 2519) with notable peaks during winter months, while individuals 65+ years recorded the lowest burden (13% of all cases, n = 1539) with sporadic infections throughout the surveillance period.

Male predominance was evident across all age groups (65% overall), with male-to-female ratios of 3.8:1 for influenza (678 males vs. 177 females), 3.3:1 for RSV (1262 males vs. 386 females), and 1.9:1 for SARS-CoV-2 (205 males vs. 110 females). This consistent pattern across diverse viral etiologies suggests healthcare access barriers rather than biological sex differences.

The temporal distribution of respiratory viruses from December 2022 to June 2025 exhibited clear seasonal patterns, with prominent peaks during the winter months (December to March) and minimal activity during the summer months (June to August). Influenza viruses, especially Influenza A (H1N1 and H3N2), and RSV were the main pathogens responsible for these peaks. The largest surge in overall viral detections occurred in January 2024, primarily driven by RSV and influenza. Winters in 2023 and 2025 showed recurring but lower peaks. SARS-CoV-2 demonstrated low to moderate circulation throughout the study period, often coinciding with influenza and RSV activity. ORVs such as hMPV, AdV, HPiV, and RV were detected at consistently low levels. Overall, the findings underscore strong winter seasonality in respiratory virus circulation, highlighting the need for ongoing surveillance during peak transmission periods (Figure 4).

### 3.2. Symptoms and Clinical Severity of RSV, SARS-CoV-2, and Influenza

The distribution of respiratory viruses across different severity indicators among the 2818 virus-positive cases.

Among patients requiring mechanical ventilation, a higher proportion were SARS-CoV-2 positive compared to those who did not need ventilation. Specifically, although SARS-CoV-2–positive individuals made up only 2.8% of the total cohort (302/10,858), they accounted for 11.4% of all patients who required mechanical ventilation (36/317), indicating a disproportionate contribution to severe respiratory support needs. Among patients, influenza-positive individuals were more likely to require ICU admission (18%), whereas only 16% needed intensive care. These results suggest that influenza was associated with more severe disease requiring ICU care, while RSV infections were generally less severe in this cohort. Patterns of HDU admission varied across different viral infections. Influenza-positive patients had a slightly higher proportion of HDU admission (18.5%) compared with RSV-positive patients. In contrast, SARS-CoV-2-positive patients were rarely admitted to HDU (0.6%). Overall, influenza was linked to slightly higher HDU admissions, RSV had minimal effect, and SARS-CoV-2 rarely required HDU support, reflecting differences in severity and management needs among these infections. A similar trend for oxygen-monitored cases can be seen in Figure 5.

These patterns indicate that while RSV and influenza drive the majority of severe SARI cases requiring intensive care, SARS-CoV-2 infections that result in hospitalization are more likely to progress to mechanical ventilation, likely reflecting disease mechanisms beyond primary respiratory tract infection.

### 3.3. Binary Logistic Regression

Logistic regression analysis identified several factors significantly linked to influenza positivity. Increasing age was associated with a slightly higher chance of influenza infection (OR = 1.00; 95% CI 1–1.01; *p* < 0.001). Male gender had 28% higher odds of infection compared to females (OR = 1.28; 95% CI 1.07–1.54; *p* = 0.01). Among symptoms, shortness of breath was strongly linked to influenza (OR = 1.62; 95% CI 1.43–1.85; *p* < 0.001), while wheezing (OR = 0.62; 95% CI 0.54–0.70; *p* < 0.001) and nasal congestion (OR = 0.60; 95% CI 0.45–0.80; *p* < 0.001) were negatively linked, indicating they were less common among influenza-positive patients. Other symptoms, including fever, malaise, headache, myalgia, cough, and sore throat, were not significantly linked to influenza infection.

For RSV, age demonstrated a significant inverse relationship with infection likelihood (OR = 0.96; 95% CI 0.96–0.96; *p* < 0.001), indicating that RSV was more common in younger individuals. Male gender (OR = 1.32; 95% CI 1.12–1.56; *p* < 0.001) was positively associated with RSV infection. Fever (OR = 0.84; 95% CI 0.72–0.97; *p* = 0.01), cough (OR = 0.77; 95% CI 0.67–0.89; *p* < 0.001), wheezing (OR = 0.79; 95% CI 0.71–0.88; *p* < 0.001), and nasal congestion (OR = 0.64; 95% CI 0.44–0.92; *p* = 0.01) were negatively associated with RSV infection, suggesting these symptoms were less common among RSV-positive cases in this dataset. Conversely, shortness of breath was positively associated with RSV (OR = 1.27; 95% CI 1.13–1.42; *p* < 0.001), indicating it was more frequent among RSV-positive patients. Other symptoms (malaise, headache, myalgia, sore throat) did not show significant associations with RSV, as shown in Table 2.

In the SARS-CoV-2 model, age showed a positive correlation with infection (OR = 1.01; 95% CI 1–1.02; *p* < 0.001), suggesting that older individuals had a higher likelihood of testing positive. Malaise was strongly linked to SARS-CoV-2 positivity (OR = 2.24; 95% CI 1.27–3.70; *p* < 0.001), as was myalgia (OR = 3.87; 95% CI 0.87–12.14; *p* = 0.04). In contrast, headache (OR = 0.48; 95% CI 0.31–0.72; *p* < 0.001) and shortness of breath (OR = 0.55; 95% CI 0.41–0.75; *p* < 0.001) were negatively associated with SARS-CoV-2 positivity.

It is important to note that the association between myalgia and SARS-CoV-2 infection, although statistically significant (*p* = 0.04), was accompanied by a wide 95% confidence interval (0.87–12.14). This interval indicates considerable uncertainty around the effect estimate and suggests that the result may be driven by limited observations within the myalgia category. The lower bound of the CI lies close to the null value, implying that the association may not be robust and could vary with small changes in the dataset. Consequently, while our findings point toward a possible relationship, the imprecision underscores the need for cautious interpretation. Larger studies with greater statistical power are warranted to confirm this association and better characterize the role of myalgia in SARS-CoV-2 infection.

Other factors like gender, fever, cough, sore throat, wheezing, and nasal congestion did not show significant associations, as illustrated in Figure 6.

All variance inflation factor (VIF) values were <5, indicating no significant multicollinearity among predictor variables. Across the three viruses, age and gender consistently emerged as key demographic predictors, although with opposite trends: older age increased the risk for influenza and SARS-CoV-2 but decreased the risk for RSV. Respiratory symptoms such as shortness of breath were positively associated with influenza and RSV, while wheezing and nasal congestion showed negative associations across multiple viruses. These distinct symptom profiles highlight the potential for clinical differentiation among these infections in mixed respiratory cases.

### 3.4. Phylogenetic Analysis and Genetic Characterization

#### 3.4.1. Influenza A(H1N1)pdm09

Phylogenetic analysis of 137 Influenza A(H1N1)pdm09 sequences (64 representative HA genes shown in the phylogenetic tree in Figure 7, analyzed with 48 global isolates) clustered within globally circulating subclades, consistent with the predominant lineages circulating worldwide during this period. All 137 sequences were classified into clades 6B.1A.5a.2a (n = 108) and 6B.1A.5a.2a.1 (n = 29) using Nextclade v3.18.1. Subclade analysis revealed sequences distributed across C.1 (n = 19), C.1.2 (n = 10), C.1.4 (n = 2), C.1.5 (n = 2), C.1.7 (n = 21), C.1.9 (n = 30), C.1.9.1 (n = 1), C.1.9.3 (n = 24), D.3 (n = 1), and D.3.1 (n = 29). Compared with the reference vaccine strain A/Wisconsin/588/2019, nucleotide divergence ranged from 0.0423% to 0.0687%, indicating less than 1% genetic variation. Several mutations in the HA gene were identified, including E29D, K54Q, D94N, T120A, R113K, N130K, P137S, K142R, K156N, I161L, I186V, A186T, D187A, A188T, I206V, T216A, E224A, A250V, R259K, T277A, and K308R.

#### 3.4.2. Influenza A(H3N2)

Phylogenetic analysis of 53 Influenza A(H3N2) sequences (phylogenetic tree in Figure 8 showing these sequences alongside global comparators, with a total of 92 HA genes analyzed) fell into globally circulating subclades, consistent with the predominant H3N2 lineages worldwide. All 53 sequences were classified into clades 3C.2a1b.2a.2b (n = 1), 3C.2a1b.2a.2a.3b (n = 10), and 3C.2a1b.2a.2a.3a.1 (n = 42) using Nextclade. Subclade analysis revealed sequences distributed across G.1.3.2 (n = 10), G.2.1 (n = 1), J (n = 3), J.1.1 (n = 2), J.2 (n = 35), and J.4 (n = 2). Nucleotide divergence from the vaccine strain A/Darwin/6/2021 ranged from 0.0249% to 0.0521%, which is less than 1%. Common mutations in the HA gene included N49S, E50K, G53N, N96S, S113A, N122D, I140M, I140K, I149M, E183K, I192F, I223V, and K276E.

#### 3.4.3. Influenza B/Victoria Lineage

Phylogenetic analysis of 27 HA genes from Pakistani samples, combined with 42 worldwide influenza B sequences, belonged to the globally circulating B/Victoria lineage. All 37 sequenced samples were classified into clades C (n = 3), C.3 (n = 9), C.5 (n = 10), C.5.6 (n = 8), and C.5.7 (n = 7) (Figure 9). Common mutations in the HA gene included I117V, H122Y, N129D, N197E, D197E, A199T, K203R, and S208P.

#### 3.4.4. SARS-CoV-2 Omicron Variants

Phylogenetic analysis was performed on 32 representative SARS-CoV-2 sequences from this study along with 76 global sequences. Pakistani sequences clustered within globally circulating Omicron subclades, confirming Pakistan’s integration into international SARS-CoV-2 transmission networks (Figure 10). All 148 SARS-CoV-2 sequences belonged to nine Omicron subclades: 23B (n = 2), 23D (n = 11), 23F (n = 2), 24A (n = 76), 24G (n = 6), 24F (n = 16), 24H (n = 7), recombinant (n = 3), 25A (n = 15), and 25C (n = 9).

Temporal analysis of the 148 sequenced Pakistani genomes revealed rapid variant succession with dominant lineages shifting every 3–4 months. The distribution analysis showed VOI (JN.1) as the most prevalent lineage at 60.14%, followed by Former VUM (XBB.1.9.1) at 6.76%, and VUM (XFG) at 6.08%. Former VOI variants XBB.1.5 and XBB.1.16 accounted for 2.70% and 2.03% respectively. More recent variants including VUM (XEC) at 11.49%, VUM (LP.8.1) at 10.14%, and Former VUM (XBB) at 0.68% were also detected (Figure 11). Early 2023 (January-April) was characterized by XBB sublineages (XBB.1.5, XBB.1.9.1, XBB.1.16). JN.1 emerged as the predominant variant by January 2024, maintaining dominance through the study period. Subsequently, newer variants including LP.8.1, XEC, and XFG lineages appeared, reflecting the continuous evolution and replacement of SARS-CoV-2 variants in the Pakistani population.

The analysis of 148 SARS-CoV-2 sequences revealed numerous mutations in the spike protein, especially in the receptor-binding domain (RBD) and N-terminal domain (NTD). Common mutations included D3H, I5T, A11T, A27S, T30A, S30P, R47K, D54N, A104V, S185L, K180R, R189S, A221D, A222V, Q229K, G252V, R345T, S486P, Q492E, S662G, P959S, V1056L, D1746Y, G1819S, P1851T, A3143V, L3829F, and T4175I.

## 4. Discussion

This integrated sentinel surveillance study establishes RSV as the predominant pathogen among hospitalized SARI patients in Pakistan, challenging the traditional emphasis on influenza in respiratory surveillance systems. The 24.6% overall detection rate aligns with regional South Asian data [19], while between the pathogen distribution RSV 58.5%, influenza 30.3%, SARS-CoV-2 11.2% of positive cases reflects the relative importance among detected viral infections. When considering all tested SARI patients, RSV accounted for 14.4%, influenza 7.5%, and SARS-CoV-2 2.8%. This demonstrates that RSV is both the predominant detected pathogen and represents substantial burden among all hospitalized SARI patients. These findings reflect both genuine epidemiologic shifts and improved detection through integrated testing protocols.

The concentration of RSV disease in infants under 2 years with substantial ICU requirements creates an immediate imperative for preventive interventions. Recent WHO recommendations support maternal RSV vaccines and nirsevimab for infant protection [20,21]. Pakistan should prioritize October-November implementation to align with December–February peak transmission. Pilot programs through existing maternal–child health platforms, monitored via this sentinel infrastructure, could demonstrate feasibility and effectiveness in South Asian settings where data remain limited. However, success requires addressing the documented healthcare access barriers for women and girls.

Phylogenetic analysis provides actionable insights for vaccine policy. While genetic divergence from current vaccine strains remains modest (<1%), accumulating substitutions in functionally critical regions warrant monitoring. H1N1pdm09 mutations in the Sa antigenic site and near glycosylation sites may affect immune recognition [22]. The H3N2 N122D substitution, linked to antigenic changes in recent 3C.2a clusters, exemplifies the rapid evolution characteristic of this subtype [23]. B/Yamagata extinction confirms WHO’s trivalent vaccine transition [24]. December–February seasonality indicates October–November vaccination timing, yet Pakistan’s low coverage demands systematic intervention: supply chain strengthening, provider training, and public education to address hesitancy.

Despite low endemic prevalence, the continued circulation and evolution of Omicron variants warrant sustained genomic surveillance. The progression through multiple Omicron subclades (23B through 25C) over the study period demonstrates ongoing viral adaptation. The progression from XBB sublineages through JN.1 to KP.2.3, LF.7.9, and XEC lineages reflects ongoing immune escape and adaptation [25]. RBD mutations enhance ACE2 binding while evading antibodies, sustaining transmission in highly immune populations [26]. Assignment to globally circulating Omicron subclades confirms Pakistan’s integration into international SARS-CoV-2 transmission networks. Unlike the strong winter seasonality observed for influenza and SARS-CoV-2 demonstrated year-round circulation throughout the study period, necessitating continuous vaccine availability and sustained genomic surveillance to detect concerning variants early.

Multivariate analysis revealed distinct symptom profiles with diagnostic utility: shortness of breath predicts influenza (OR = 1.62) and RSV (OR = 1.27), while systemic symptoms indicate SARS-CoV-2 (malaise OR = 2.24, myalgia OR = 3.87). However, substantial overlap limits symptom-based diagnosis reliability, reinforcing laboratory confirmation needs. The disproportionate mechanical ventilation requirement among hospitalized SARS-CoV-2 patients 6.1% of ventilation cases versus 2.8% disease burden suggests distinct pathophysiology beyond primary respiratory infection, possibly reflecting endothelial involvement and immunopathology not seen with influenza or RSV.

The 65–79% male predominance across all pathogens represents a critical equity failure, almost certainly reflecting socio-cultural healthcare barriers rather than biology given consistency across diverse viral etiologies. Pakistani women face intersecting obstacles: limited decision-making autonomy, mobility restrictions, and lower prioritization for healthcare expenditure. This pattern indicates severe underestimation of true female disease burden. Addressing this requires multi-level intervention: female community health workers, subsidized care, flexible services, and male engagement in health education. Future surveillance must implement targeted female enrollment strategies and community-based case finding to separate biological from social determinants. Any prevention program RSV immunization, influenza vaccination will fail to achieve population impact without addressing these fundamental access barriers.

Several limitations affect interpretation. Urban tertiary hospital focus limits rural generalizability. SARI case definition captures severe disease but misses the community burden spectrum. Selective multiplex testing (10% of negatives, prioritizing severe cases) introduces substantial ascertainment bias for additional pathogens prevalence estimates are unreliable and ICU rates misleading since testing targeted the sickest patients. Universal multiplex testing in future cycles would clarify whether rhinovirus, metapneumovirus, adenovirus, and parainfluenza contribute substantially to SARI or primarily cause mild disease [27]. Absence of bacterial testing prevents co-infection assessment, which influences severity and treatment. Sequencing only high viral load samples (Ct < 28–30) may bias toward severe infections. Lack of systematic vaccination data precludes test-negative vaccine effectiveness studies despite this being a key sentinel surveillance objective.

The surveillance infrastructure established trained personnel at seven sites, validated protocols, sequencing capacity, and GIHSN integration, representing a strategic investment that extends beyond research. Sustaining this requires institutional commitment and integration into routine public health functions rather than project-dependent funding. Immediate priorities include implementing RSV prevention pilots in October 2025, expanding influenza vaccination through systematic barrier reduction, maintaining SARS-CoV-2 genomic surveillance due to rapid evolution, and enhancing December–February surge preparedness through capacity building and resource prepositioning. Universal multiplex testing and bacterial pathogen inclusion would strengthen etiologic characterization. Most critically, addressing gender disparities must be central to all interventions, as equity is not optional.

Respiratory viruses impose a significant SARI burden in Pakistan with distinct age, seasonal, and clinical patterns. Evidence-based interventions like RSV prevention, influenza vaccination expansion, and continued SARS-CoV-2 surveillance can reduce disease impact, but only if healthcare access barriers are systematically addressed. The established surveillance infrastructure positions Pakistan to contribute meaningfully to regional and global health security while supporting national priorities. Sustained investment in surveillance systems, equity-focused implementation, and active global data sharing will be essential for protecting vulnerable populations and enhancing pandemic preparedness.

## Figures and Tables

**Figure 1 viruses-18-00026-f001:**
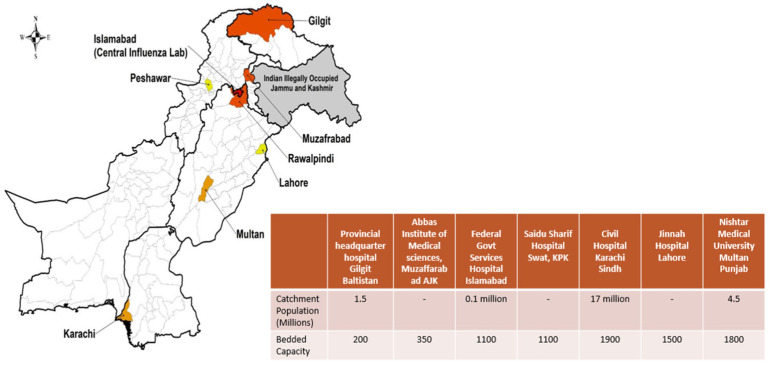
GIHSN Respiratory Viruses Sentinel Surveillance Network: Public Tertiary Care Hospitals in Pakistan.

**Figure 2 viruses-18-00026-f002:**
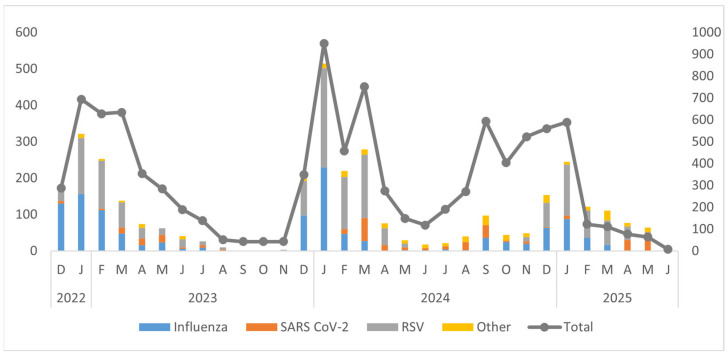
Annual trends of different viruses in Pakistan by month and year (December 2022–June 2025).

**Figure 3 viruses-18-00026-f003:**
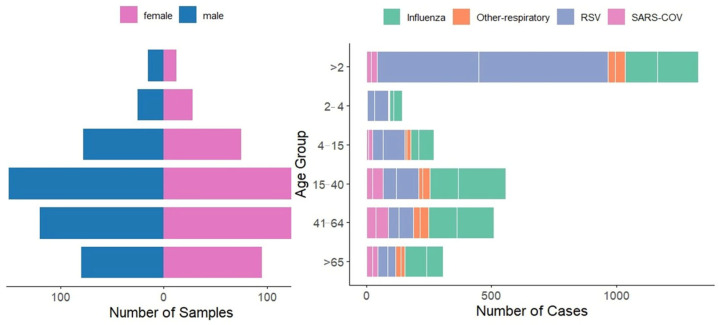
Gender and Age Group Distribution of Respiratory Virus Cases (Influenza, RSV, SARS-CoV-2, and Other Respiratory Viruses).

**Figure 4 viruses-18-00026-f004:**
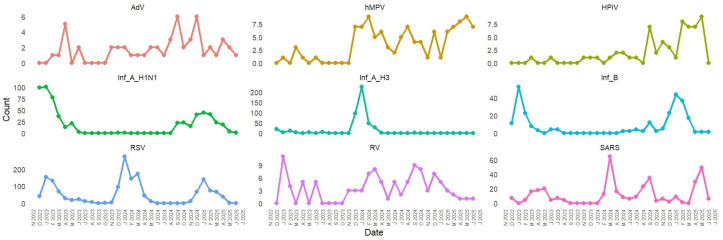
Temporal Distribution of Respiratory Viruses (RSV, Influenza, SARS-CoV-2, and Other Pathogens) from 2022 to 2025.

**Figure 5 viruses-18-00026-f005:**
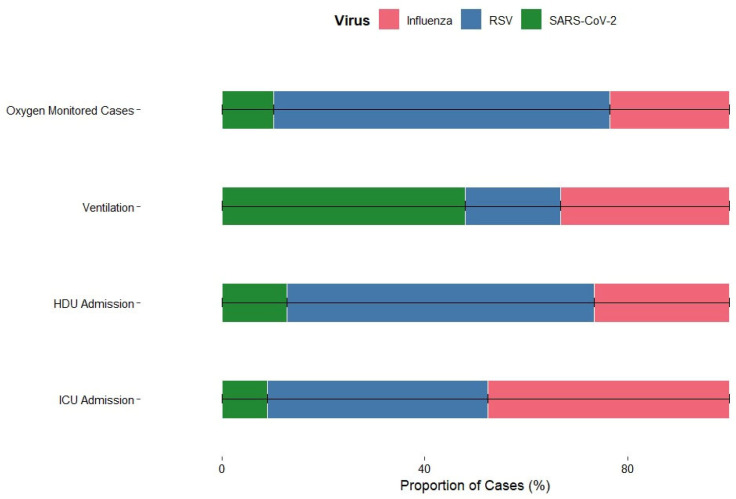
Severity Profile of Influenza, RSV, and SARS-CoV-2 Based on Clinical Outcomes.

**Figure 6 viruses-18-00026-f006:**
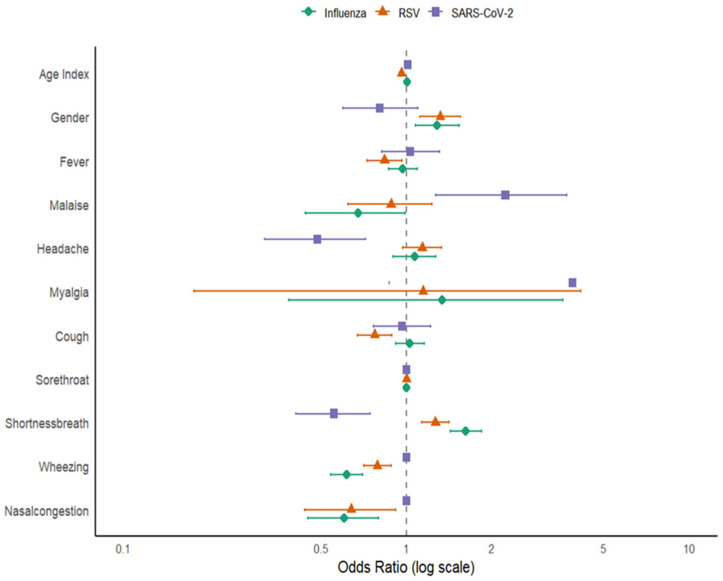
Forest plot for Factors Associated with Influenza, RSV, and SARS-CoV-2 Infections.

**Figure 7 viruses-18-00026-f007:**
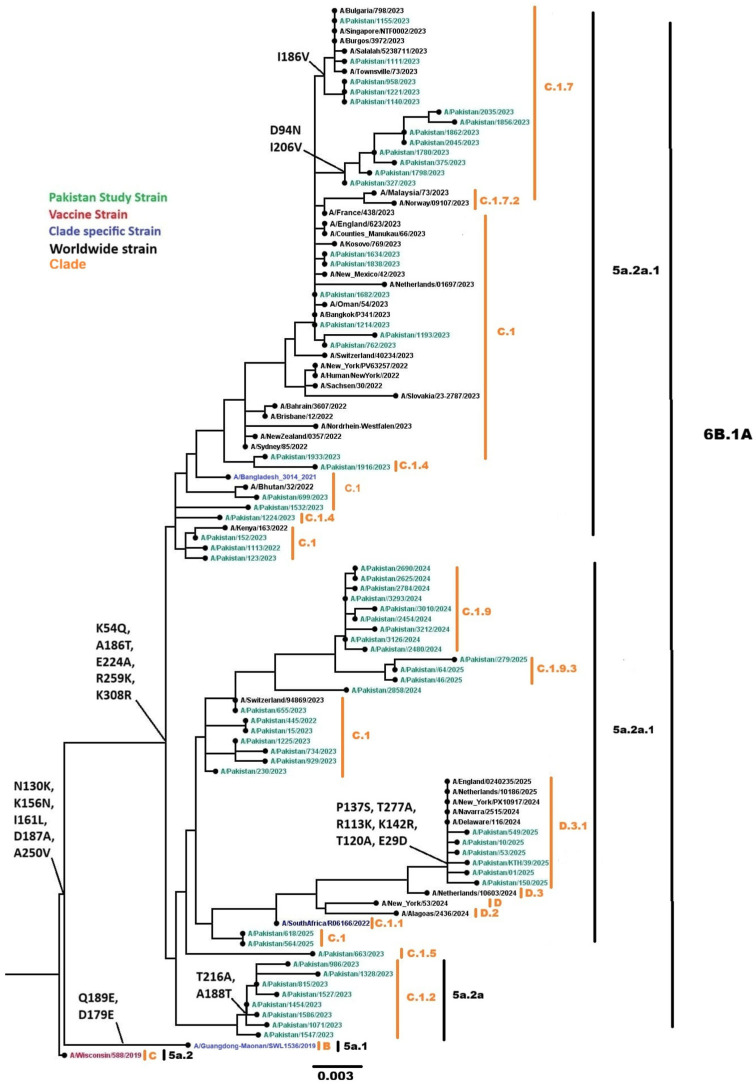
Phylogenetic tree of the Influenza A (H1N1) pdm09 HA gene. Worldwide sequences are in black, with the vaccine strain highlighted in red, study sequences in green, and subclade-specific reference strains in blue.

**Figure 8 viruses-18-00026-f008:**
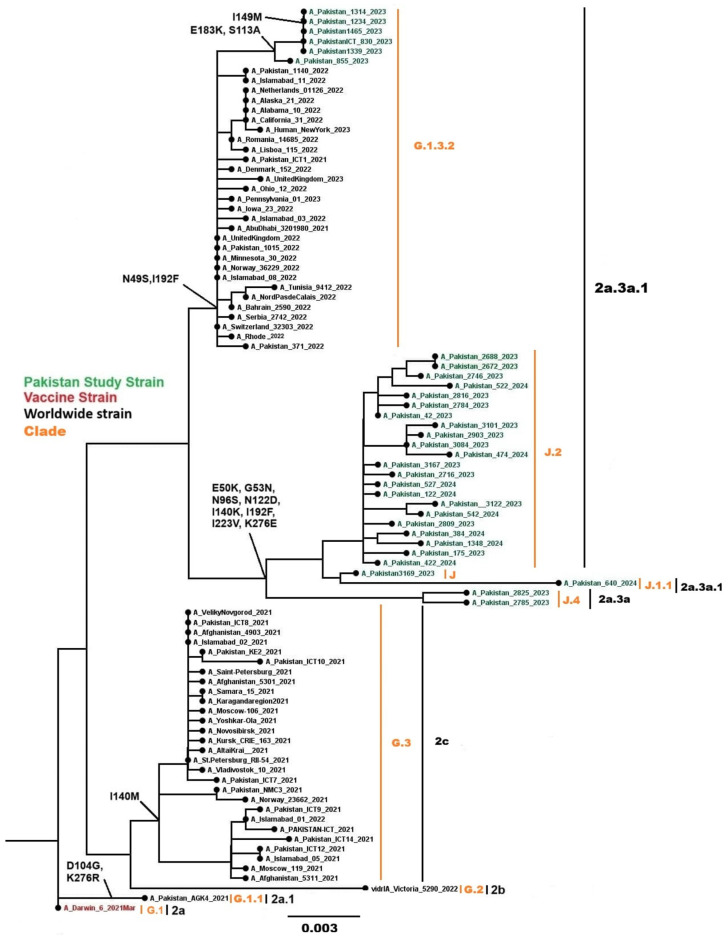
Phylogenetic tree of the Influenza A (H3N2) HA gene. Worldwide sequences are in black, with the vaccine strain highlighted in red and Pakistan study sequences in green.

**Figure 9 viruses-18-00026-f009:**
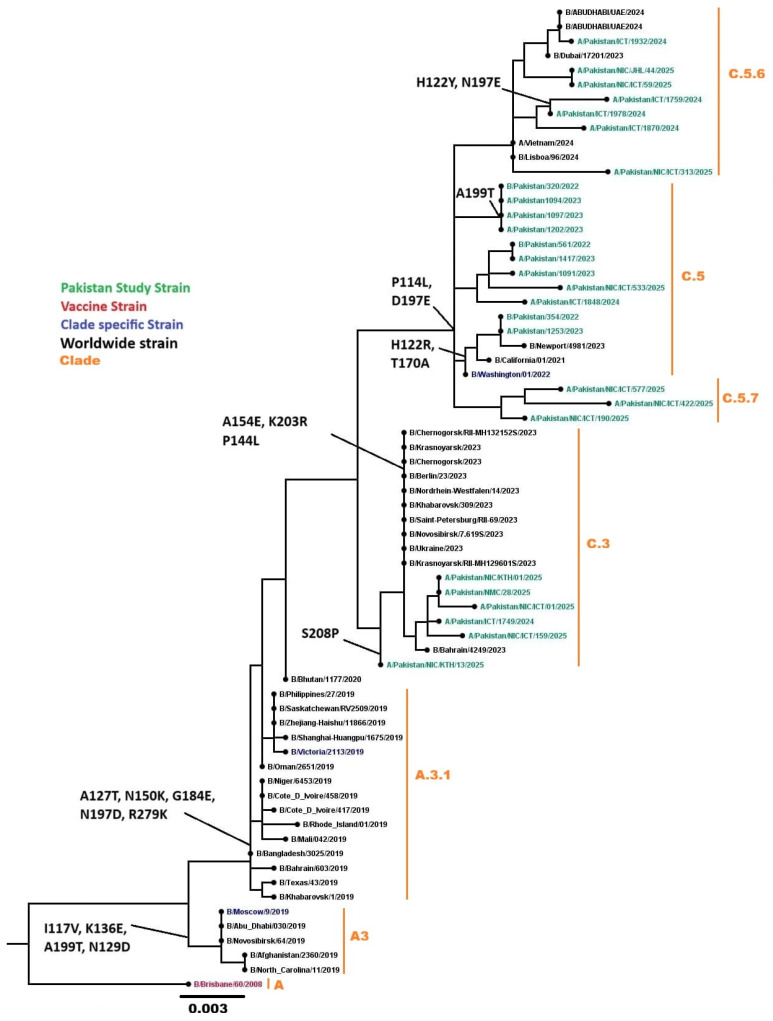
Phylogenetic analysis of the Influenza B HA gene. Worldwide sequences are in black, with the vaccine strain highlighted in red, Pakistan study sequences in green, and subclades-specific reference strains in blue.

**Figure 10 viruses-18-00026-f010:**
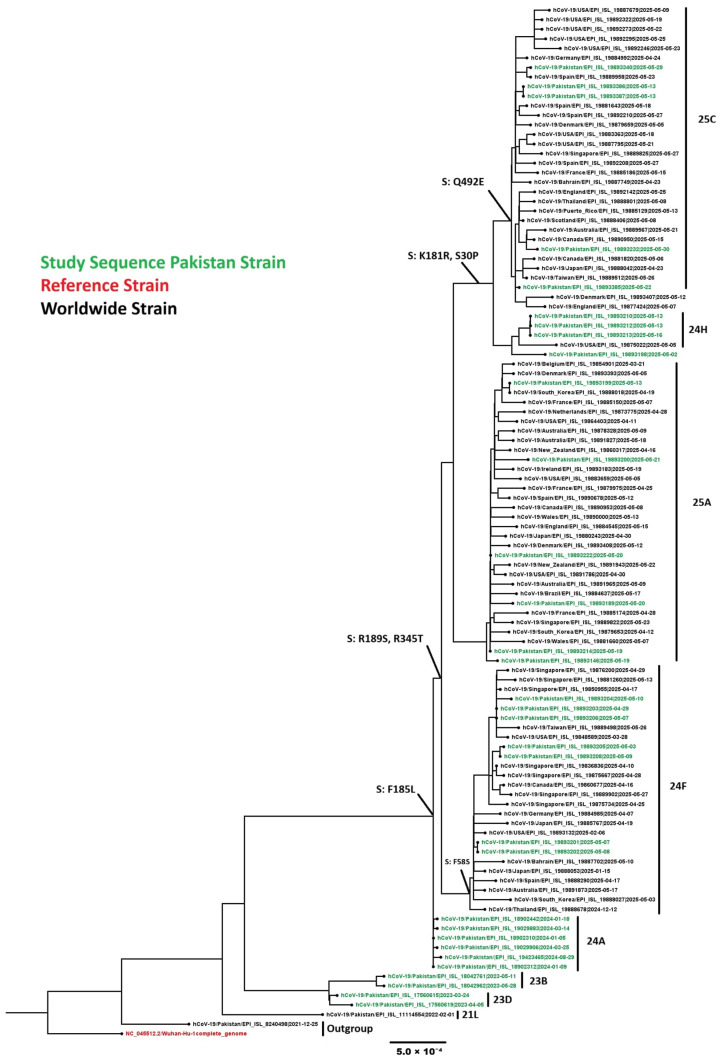
Phylogenetic analysis of the SARS-CoV-2 sequences. Worldwide sequences are in black, with the vaccine strain highlighted in red and Pakistan sequences in green.

**Figure 11 viruses-18-00026-f011:**
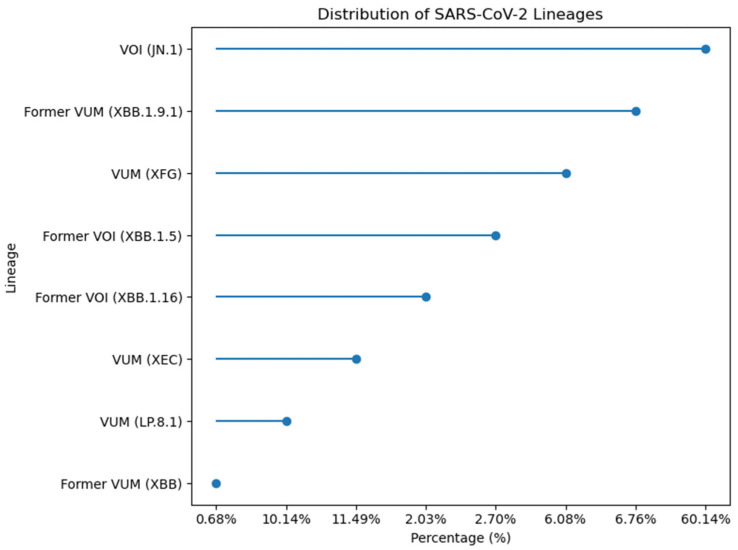
Percentage distribution of SARS-CoV-2 lineages identified in 148 sequenced genomes from Pakistan.

**Table 1 viruses-18-00026-t001:** Epidemiological and clinical characteristics.

Characteristic	Total Cases n (%)	Influenza n (%)	Influenza *p*-Value	SARS-CoV-2 n (%)	SARS-CoV-2 *p*-Value	RSV n (%)	RSV *p*-Value
Total Tested	11,451	855 (7.5)		315 (2.8)		1648 (14.4)	
Male	7486 (65)	678 (79.3)	<0.001	205 (65.1)	0.157	1262 (76.6)	<0.001
Female	3965 (35)	177 (20.7)		110 (34.9)		386 (23.4)	
Age Group (years)							
<2	3312 (29)	293 (34.3)	<0.001	42 (13.3)	<0.001	1039 (63.0)	<0.001
2–4	459 (4)	50 (5.8)		3 (1.0)		85 (5.2)	
5–14	873 (8)	92 (10.8)		24 (7.6)		130 (7.9)	
15–40	2749 (24)	302 (35.3)		68 (21.6)		140 (8.5)	
41–64	2519 (22)	261 (30.5)		87 (27.6)		102 (6.2)	
≥65	1539 (13)	154 (18.0)		46 (14.6)		72 (4.4)	
Fever	4521 (39.5)	485 (56.7)	0.003	228 (72.4)	0.558	995 (60.4)	<0.001
Cough	5725 (50.0)	472 (55.2)	0.008	189 (60.0)	0.498	679 (41.2)	<0.001
Sore throat	4922 (43.0)	449 (52.5)	<0.001	131 (41.6)	0.009	524 (31.8)	0.009
Shortness of breath	3600 (31.4)	476 (55.7)	<0.001	79 (25.1)	0.001	682 (41.4)	<0.001
Wheezing	2865 (25.0)	198 (23.2)	0.024	45 (14.3)	0.089	512 (31.1)	<0.001
Nasal congestion	1832 (16.0)	89 (10.4)	<0.001	38 (12.1)	0.156	298 (18.1)	0.01
Malaise	1374 (12.0)	87 (10.2)	0.06	89 (28.3)	<0.001	156 (9.5)	0.47
Headache	2290 (20.0)	189 (22.1)	0.45	32 (10.2)	<0.001	267 (16.2)	0.11
Myalgia	687 (6.0)	48 (5.6)	0.6	67 (21.3)	0.04	89 (5.4)	0.86
Chest Pain	344 (3.0)	43 (5.0)	0.433	20 (6.3)	<0.001	28 (1.7)	0.089
Cardiovascular disease	344 (3.0)	23 (2.7)	0.569	20 (6.3)	0.031	13 (0.8)	<0.001
Asthma	229 (2.0)	29 (3.4)	0.06	3 (1.0)	1	20 (1.2)	0.299
COPD	687 (6.0)	63 (7.4)	0.812	20 (6.3)	0.41	50 (3.0)	<0.001
Diabetes	229 (2.0)	14 (1.6)	0.799	4 (1.3)	0.699	17 (1.0)	0.57
Renal disease	115 (1.0)	2 (0.2)	0.418	3 (1.0)	0.484	1 (0.1)	0.02
Liver Cirrhosis	57 (0.5)	1 (0.1)	0.525	1 (0.3)	1	0 (0)	0.59
Influenza Vaccinated	456 (4.0)	22 (2.6)	-	-	-	-	-
SARS-CoV-2 Vaccinated	4439 (38.8)	-	-	164 (52.1)	-	-	-

**Note:** *p*-value for Age Groups calculated using chi-square test of independence, comparing the distribution across all age categories. *p*-value represents an overall association and not pairwise comparisons.

**Table 2 viruses-18-00026-t002:** Binary logistic regression summary statistic.

Virus	Covariate	Odds Ratio	95% CI	*p*-Value
Influenza	Age (years)	1	1.00–1.01	<0.001
	Male gender	1.28	1.07–1.54	0.01
	Fever	0.97	0.86–1.09	0.63
	Malaise	0.67	0.44–0.99	0.06
	Headache	1.07	0.90–1.27	0.45
	Myalgia	1.33	0.38–3.59	0.6
	Cough	1.03	0.91–1.16	0.64
	Sore throat	1	0.99–1.00	0.77
	Shortness of breath	1.62	1.43–1.85	<0.001
	Wheezing	0.62	0.54–0.70	<0.001
	Nasal congestion	0.6	0.45–0.80	<0.001
RSV	Age (years)	0.96	0.96–0.96	<0.001
	Male gender	1.32	1.12–1.56	<0.001
	Fever	0.84	0.72–0.97	0.01
	Malaise	0.88	0.62–1.23	0.47
	Headache	1.14	0.97–1.33	0.11
	Myalgia	1.14	0.18–4.14	0.86
	Cough	0.77	0.67–0.89	<0.001
	Sore throat	1	0.99–1.00	0.86
	Shortness of breath	1.27	1.13–1.42	<0.001
	Wheezing	0.79	0.71–0.88	<0.001
	Nasal congestion	0.64	0.44–0.92	0.01
SARS-CoV-2	Age (years)	1.01	1.00–1.02	<0.001
	Male gender	0.8	0.60–1.10	0.16
	Fever	1.03	0.82–1.31	0.78
	Malaise	2.24	1.27–3.70	<0.001
	Headache	0.48	0.31–0.72	<0.001
	Myalgia	3.87	0.87–12.14	0.04
	Cough	0.97	0.76–1.22	0.77
	Sore throat	1	1.00–1.00	0.64
	Shortness of breath	0.55	0.41–0.75	<0.001
	Wheezing	1	0.99–1.00	0.48
	Nasal congestion	1	0.99–1.00	0.83

## Data Availability

All relevant data are within the paper tables and figure files. All influenza and SARS-CoV-2 sequences generated in this study have been submitted to the GISAID) database. Individual GISAID accession numbers for all 375 sequences are provided in Appendix A.

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
