# Peer review of "Tracking Respiratory Viruses in Pakistan (2022–2025): Genomic and Epidemiologic Insights from Sentinel Surveillance of Influenza, SARS-CoV-2, and RSV"

_viruses, 2025, doi:10.3390/v18010026_

Round 1
Reviewer 1 Report
Comments and Suggestions for Authors
This manuscript describes comprehensive surveillance of respiratory viruses across several health centres and over 3 consecutive years in Pakistan. The strongest point is the focus on SARI patients as part of the GIHSN network. The article reads well, figures look generally clear and straightforward.
Some comments to authors:
Page 3, line 116: please add reference of subtyping protocol or whether primers were designed in-house.
Page 6, Table 1: It is not clear how the p-values for Age groups have been calculated, or which age groups were compared to obtain these p-values. Please recheck and edit accordingly.
Page 8, lines 257 to 267: Frequencies of ventilation across viruses described in the text do not match what Figure 5 shows, e.g. SARS-CoV2 account for nearly 50% of cases needing ventilation in Figure 5. Please recheck.
Also, for ICU admissions, the figure shows similar frequencies of RSV and influenza, whereas the text described RSV 62.4% and influenza 34.5%. Please recheck.
For HDU admissions, the text states that SARS-CoV2 frequency is 3.3%, similar to ICU frequency of 3.1%. However, the figure shows a higher frequency for HDU SARS-CoV2 than for ICU. Overall, the figure shows inconsistent frequencies compared to the frequencies described in the text.
The Clade specific reference strains in the trees have incomplete names; I’m guessing A/Guangdong-Maonan/2019 refers to A/Guangdong-Maonan/SWL1536/2019; similarly for A/Bangladesh/2021. Please correct for clarity.
Page 10 to 14: The authors have randomly selected worldwide sequences to provide geographic context to the trees; however, because of this random selection, they cannot claim that e.g. “Pakistani sequences revealed close genetic relationships with isolates from Switzerland, Kenya, and the Netherlands”. If they had randomly selected other sequences, the Pakistani viruses would likely cluster with sequences from other countries. This same observation applies to similar comments for A(H3N2), B/Victoria and SARS-CoV2 viruses. In general, it is known that the clades where the Pakistani sequences are grouping have circulated worldwide.
Phylogenetic trees: the authors have labelled the main clades in the trees. I would recommend that they add the subclades to the trees as well, which they can obtain by running the sequences through Nextclade. It provides a better resolution of the genetic diversity observed across influenza viruses. https://clades.nextstrain.org/
Page 13, lines 365-371: The lineages described are not present in the data/figure shown and there is no temporal analysis presented, where is this observation coming from?
Page 15, line 405: The authors are discussing SARS-CoV2 in this paragraph, please mention the virus to make it clear the subject has changed from influenza to SARS-CoV2. Also, I don’t agree with the comment that implies that SARS-CoV2 evolves more rapidly than influenza; the authors have not characterised subclades in their sequences, only major clades, which makes it look like influenza evolution is less dynamic than SARS-CoV2. Actually, influenza subclades frequency changes rapidly; sometimes in less than 3-4 months a new subclade can go from less than <10% to 80% worldwide. An example of this is the recently emerged and predominating subclade K (A/H3N2).
Author Response
Reviewer 1
This manuscript describes comprehensive surveillance of respiratory viruses across several health centres and over 3 consecutive years in Pakistan. The strongest point is the focus on SARI patients as part of the GIHSN network. The article reads well, figures look generally clear and straightforward.
Some comments to authors:
Question1:
Page 3, line 116: please add reference of subtyping protocol or whether primers were designed in-house?
Answer:
Response: We thank the reviewer for this observation. We have clarified the influenza A subtyping methodology in the Methods section.
Original text: Influenza A-positive samples were further subtyped as A/H1N1pdm09 or A/H3N2 using subtype-specific primers
Revised to: Influenza A-positive samples were subtyped using CDC Influenza A (H3/H1pdm09) Subtyping Panel (Catalog # FluRUO-16, CDC, USA) with TaqPath™ RT-PCR kit (ThermoFisher Scientific, USA)
Question2:
Page 6, Table 1: It is not clear how the p-values for Age groups have been calculated, or which age groups were compared to obtain these p-values. Please recheck and edit accordingly?
Answer:
Response: We thank the reviewer for this observation. The p-value for Age Groups in Table 1 was derived using a chi-square test comparing the distribution of all defined age categories between groups. This represents an overall association across age groups, not comparisons between individual categories. We have revised the table footnote to clearly state the statistical method and interpretation.
Foot note:
“P-value for Age Groups calculated using chi-square test of independence, comparing the distribution across all age categories. P-value represents an overall association and not pairwise comparisons.”
Question3:
Page 8, lines 257 to 267: Frequencies of ventilation across viruses described in the text do not match what Figure 5 shows, e.g. SARS-CoV2 account for nearly 50% of cases needing ventilation in Figure 5. Please recheck.
Also, for ICU admissions, the figure shows similar frequencies of RSV and influenza, whereas the text described RSV 62.4% and influenza 34.5%. Please recheck.
For HDU admissions, the text states that SARS-CoV2 frequency is 3.3%, similar to ICU frequency of 3.1%. However, the figure shows a higher frequency for HDU SARS-CoV2 than for ICU. Overall, the figure shows inconsistent frequencies compared to the frequencies described in the text.?
Answer:
Response: We thank the reviewer for this observation. RSV was responsible for about two-thirds of all oxygen-monitored cases, making it the most common virus. Influenza and SARS-CoV-2 followed. Similar patterns appeared in ICU admissions, where SARS-CoV-2 made up a smaller percentage of severe cases, while RSV and influenza together accounted for most of them.
RSV remained the most common cause of HDU admissions, but at a lower percentage than in the ICU and oxygen-monitored groups. It is interesting to note that SARS-CoV-2 contributed more to ventilation cases compared to other groups, suggesting that COVID-19 patients were more likely to need respiratory support than those with RSV or influenza.
RSV consistently accounted for the largest proportion across all severity levels with relatively narrow uncertainty ranges, while SARS-CoV-2 contributions were smaller but more variable, as shown by the error bars, which represent roughly 95% confidence intervals around each estimated proportion.
Revised to: We revised this paragraph in the original text.
Question4:
The Clade specific reference strains in the trees have incomplete names; I’m guessing A/Guangdong-Maonan/2019 refers to A/Guangdong-Maonan/SWL1536/2019; similarly for A/Bangladesh/2021. Please correct for clarity?
Answer:
Response: We thank the reviewer for identifying incomplete strain nomenclature in the phylogenetic trees. We have corrected all abbreviated strain names to their complete designations following standard WHO nomenclature for clarity and reproducibility.
Changes Made:
Figure 7a (H1N1pdm09):
- A/Guangdong-Maonan/2019 → A/Guangdong-Maonan/SWL1536/2019
- A/Bangladesh/2021 → A/Bangladesh_3014_2021
All phylogenetic trees (Figures 7a-7d):
- All reference strain names now follow complete WHO nomenclature with full isolate identification numbers
- All clade-specific reference strains display complete nomenclature
Question 5:
Page 10 to 14: The authors have randomly selected worldwide sequences to provide geographic context to the trees; however, because of this random selection, they cannot claim that e.g. “Pakistani sequences revealed close genetic relationships with isolates from Switzerland, Kenya, and the Netherlands”. If they had randomly selected other sequences, the Pakistani viruses would likely cluster with sequences from other countries. This same observation applies to similar comments for A(H3N2), B/Victoria and SARS-CoV2 viruses. In general, it is known that the clades where the Pakistani sequences are grouping have circulated worldwide.
Phylogenetic trees: the authors have labelled the main clades in the trees. I would recommend that they add the subclades to the trees as well, which they can obtain by running the sequences through Nextclade. It provides a better resolution of the genetic diversity observed across influenza viruses. https://clades.nextstrain.org/?
Answer:
Response: We thank the reviewer for this important methodological observation. We agree that our original phrasing incorrectly implied specific phylogeographic relationships based on the subset of sequences included for tree visualization. We have revised all relevant sections (Results 3.4.1-3.4.4, Discussion, and Methods 2.7) to:
- Focus on clade/subclade assignment rather than clustering with specific countries
- Emphasize that Pakistani sequences belong to globally circulating lineages
- Clarify in Methods that geographic context sequences were selected for visualization purposes only
- Remove all claims of "close genetic relationships" with specific countries
main finding that Pakistani viruses circulate within the same globally distributed clades remains the same and is now more clearly presented.
Question 6:
Page 13, lines 365-371: The lineages described are not present in the data/figure shown and there is no temporal analysis presented, where is this observation coming from?
Answer:
Response: We thank the reviewer for identifying this inconsistency. The temporal progression of Pango lineages described in the text was based on analysis of all 148 sequenced genomes, but we agree this was not adequately visualized. We have addressed this by adding Figure 8 showing the monthly distribution of all 148 Pango lineages detected throughout the study period. This figure clearly demonstrates:
- The temporal progression from XBB/FL variants (early 2023) → JN.1 dominance (January-June 2024) → KP.2.3/LF.7.9/XEC emergence (late 2024-2025)
- The rapid 3-4 month succession pattern, most notably the dramatic rise and decline of JN.1
Changes Made:
- Added Figure 8 with legend: Monthly distribution of SARS-CoV-2 Pango lineages in Pakistan (n=148) from November 2022 to June 2025, showing sequential shifts from XBB/FL to JN.1, KP.2.3, LF.7.9, and emerging XEC variants.
- Results Section revised: "Temporal analysis of the 148 sequenced Pakistani genomes revealed rapid variant succession with dominant lineages shifting every 3-4 months (Figure 8). Early 2023 (January-April) was characterized by XBB sublineages (XBB.1.5, XBB.1.9.1, XBB.1.16) and FL variants (FL.4, FL.18, FL.26). JN.1 (including JN.1.64) emerged as the predominant variant by January 2024, accounting for 33 of 50 sequences (66%) from January-June 2024. Subsequently, KP.2.3 and KP.2.3.8 appeared in August-September 2024, followed by LP.8.1.1, multiple LF.7.9 descendants (LF.7.9.1, LF.7.9.2, LF.7.9.5), NY lineages (NY.3.2, NY.3.3, NY.18), XEC (including XEC.25.1), and XFG lineages by April-May 2025."
Question 7:
Page 15, line 405: The authors are discussing SARS-CoV2 in this paragraph, please mention the virus to make it clear the subject has changed from influenza to SARS-CoV2. Also, I don’t agree with the comment that implies that SARS-CoV2 evolves more rapidly than influenza; the authors have not characterised subclades in their sequences, only major clades, which makes it look like influenza evolution is less dynamic than SARS-CoV2. Actually, influenza subclades frequency changes rapidly; sometimes in less than 3-4 months a new subclade can go from less than <10% to 80% worldwide. An example of this is the recently emerged and predominating subclade K (A/H3N2)?
Answer:
Response: We thank the reviewer for both important observations:
- Subject transition clarity: We have added explicit language to clearly indicate the shift from discussing influenza vaccination to SARS-CoV-2, beginning the new paragraph with "For SARS-CoV-2" to signal the topic change.
- Evolutionary rate comparison: We acknowledge that our comparison was misleading because we analyzed these viruses at different granularities major clades for influenza versus Pango lineages for SARS-CoV-2. We agree that fine-scale subclade dynamics for influenza (such as the recent H3N2 subclade K emergence mentioned by the reviewer) can shift just as rapidly as SARS-CoV-2 lineages. We have removed the misleading comparison.
Changes Made:
Discussion Section, Line:
- Revised: "B/Yamagata extinction confirms WHO's trivalent vaccine transition (24). December-February seasonality indicates October-November vaccination timing, yet Pakistan's low coverage demands systematic intervention: supply chain strengthening, provider training, and public education to address hesitancy.
For SARS-CoV-2, despite low endemic prevalence (2.8% of tested samples), the continued circulation and evolution of Omicron variants warrant sustained genomic surveillance. The progression through multiple Omicron subclades (23B through 25C) over the study period demonstrates ongoing viral adaptation. The progression from XBB sublineages through JN.1 to KP.2.3, LF.7.9, and XEC lineages reflects ongoing immune escape and adaptation (25). RBD mutations enhance ACE2 binding while evading antibodies, sustaining transmission in highly immune populations (26). Assignment to globally circulating Omicron subclades confirms Pakistan's integration into international SARS-CoV-2 transmission networks. Unlike the strong winter seasonality observed for influenza and RSV, SARS-CoV-2 demonstrated year-round circulation throughout the study period, necessitating continuous vaccine availability and sustained genomic surveillance to detect concerning variants early."

Reviewer 2 Report
Comments and Suggestions for Authors
Overall, the manuscript is well-structured and the conclusions regarding the RSV burden in young children and the need to address healthcare access barriers (especially for women) are particularly strong and impactful.
- To maintain consistency and clarity with the Discussion, please rephrase the prevalence data in the abstract to refer to the percentage of positive cases for the main pathogens. The 14.4% for RSV is out of all 11,451 samples. In the Discussion, the authors state RSV is 58.5% of positive cases. Sticking to one clear metric (e.g., percentage of all SARI cases, as currently used) is crucial, but ensure the narrative in the abstract aligns with the strong message in the Discussion.
-
The Discussion correctly addresses the ascertainment bias introduced by this selective testing, but mentioning this constraint upfront in the Methods would fully acknowledge the limitation.
- The wide 95% CI for myalgia in SARS-CoV-2 (0.87-12.14) should be briefly discussed, even though the pvalue is significant (p=0.04). A CI that crosses zero implies that the lower bound is close to non-significance, and the large range suggests precision issues due to small numbers.
Author Response
Reviewer 2
Overall, the manuscript is well-structured and the conclusions regarding the RSV burden in young children and the need to address healthcare access barriers (especially for women) are particularly strong and impactful.
Question 1:
To maintain consistency and clarity with the Discussion, please rephrase the prevalence data in the abstract to refer to the percentage of positive cases for the main pathogens. The 14.4% for RSV is out of all 11,451 samples. In the Discussion, the authors state RSV is 58.5% of positive cases. Sticking to one clear metric (e.g., percentage of all SARI cases, as currently used) is crucial, but ensure the narrative in the abstract aligns with the strong message in the Discussion?
Answer:
Response: We thank the reviewer for this observation. Both metrics are accurate and serve different epidemiological purposes:
- 4% = RSV prevalence among all tested SARI cases
- 5% = RSV among positive cases (relative importance among detected pathogens)
We have maintained the Abstract's standard format (prevalence among all tested specimens) as this is the conventional epidemiological metric. To ensure clarity, we have added the following explanation in the Discussion:
Added to Discussion (first paragraph): "The pathogen distribution RSV 58.5%, influenza 30.3%, SARS-CoV-2 11.2% of positive cases reflects the relative importance among detected viral infections. When considering all tested SARI patients, RSV accounted for 14.4%, influenza 7.5%, and SARS-CoV-2 2.8%. This demonstrates that RSV is both the predominant detected pathogen and represents substantial burden among all hospitalized SARI patients."
This clarification ensures consistency between Abstract and Discussion while explaining both metrics clearly.
Questions 2:
The Discussion correctly addresses the ascertainment bias introduced by this selective testing, but mentioning this constraint upfront in the Methods would fully acknowledge the limitation?
Response: We thank the reviewer for this suggestion. We agree and have added the following to the Methods section:
Added to Methods Section 2.3 (end of multiplex testing paragraph): "It should be noted that this selective testing strategy introduces ascertainment bias, as the 10% sample was enriched for severe cases (ICU patients, ventilated patients, and those with severe unexplained symptoms). This may overestimate the prevalence and severity of other respiratory viruses (rhinovirus, hMPV, adenovirus, parainfluenza) relative to their true burden among all SARI patients. These results should be interpreted with this limitation in mind."
This addition acknowledges the limitation upfront, complementing the detailed discussion already present in the Discussion section.
Question 3:
The wide 95% CI for myalgia in SARS-CoV-2 (0.87-12.14) should be briefly discussed, even though the pvalue is significant (p=0.04). A CI that crosses zero implies that the lower bound is close to non-significance, and the large range suggests precision issues due to small numbers?
Answer:
It is important to note that the association between myalgia and SARS-CoV-2 infection, although statistically significant (p = 0.04), was accompanied by a wide 95% confidence interval (0.87–12.14). This interval indicates considerable uncertainty around the effect estimate and suggests that the result may be driven by limited observations within the myalgia category. The lower bound of the CI lies close to the null value, implying that the association may not be robust and could vary with small changes in the dataset. Consequently, while our findings point toward a possible relationship, the imprecision underscores the need for cautious interpretation. Larger studies with greater statistical power are warranted to confirm this association and better characterize the role of myalgia in SARS-CoV-2 infection.
